# Validation of Lithuanian Arthroplasty Register Telephone Survey of 2769 Patients Operated for Total Knee Replacement

**DOI:** 10.3390/medicina55060310

**Published:** 2019-06-25

**Authors:** Egle Terteliene, Kazimieras Grigaitis, Otto Robertsson, Narunas Porvaneckas, Jolanta Dadoniene, Algirdas Venalis

**Affiliations:** 1Faculty of Medicine, Vilnius University, M.K. Čiurlionio 21/27, LT-03101 Vilnius, Lithuania; narunas.porvaneckas@mf.vu.lt (N.P.); jolanta.dadoniene@mf.vu.lt (J.D.); algirdas.venalis@mf.vu.lt (A.V.); 2The Department of Orthopedics, Medical Academy, Lithuanian University of Health Sciences, Eivenių 2, LT- 50161 Kaunas, Lithuania; grigaitis.kazimieras@gmail.com; 3Department of Clinical Sciences and Department of Orthopaedics, Lund University and Lund University Hospital, Sölvegatan 19 - BMC F12 Lund, Sweden; otto.robertsson@med.lu.se; 4State Research Institute Center for Innovative Medicine, LT–08410 Vilnius, Lithuania

**Keywords:** arthroplasty, register, knee, revision, interview

## Abstract

*Background and objectives*: The aim of our study is to validate the registration of knee arthroplasty revisions in the Lithuanian Arthroplasty Register (LAR) and thus give an indication of the accuracy of the published revision rates. *Materials and methods*: A total of 4269 primary total knee arthroplasties (TKAs) registered in the LAR between 2013 and 2015 were included. Two years after surgery the patients were contacted by phone in order to inquire if they had been subject to revision. The information from the patients was then cross checked against what had been registered in the LAR, and in case of a revision not having been registered hospital charts were investigated. Thus, the patients were followed up with regarding revision and/or death until 2017. A true revision was defined as an addition, exchange, or removal of one or all components. *Results*: Out of 4269 primary TKAs, we managed to contact and interview 2769 patients. Nine small hospitals were not able to provide contact details (telephone numbers) for 533 patients (549 knees). Sixty-seven patients (67 knees) were deceased (data from the Lithuanian National Census Register) and a further 438 patients (565 knees) appeared to have a wrong or non-valid telephone number, leaving 3031 (3091 knees) patients being contacted. Of those, 262 patients (266 knees) refused to participate in the study which left 2769 responders (2825 knees). Sixty-one patients said that reoperation had been performed on the index knee within two years of their primary surgery. After checking with the clinics, 10 were surgical procedures on the knee but not true revisions by our criteria. Out of the 51 true revisions we found that 46 were registered to the LAR as revised, while five (9.8%) revisions were missing. *Conclusions*: We conclude that the Lithuanian Arthroplasty Register has a good completeness of registered revision TKAs as only 9.8% of revisions were missing.

## 1. Introduction

National arthroplasty registers are valuable tools, monitoring the outcomes of joint arthroplasties by evaluating their risk of revision [1,2,3]. For the analyses to be relevant and useful for decision making, the register needs to have reasonable completeness and unbiased reporting of primaries, in combination with a high completeness in reporting on those primaries becoming subject to revision. In order to ensure and document that this is the case, the register information needs to be validated [4,5].

Three common methods are used to validate the data quality and completeness of arthroplasty registers. Comparisons of national registry data with (A) data from national patient administrative systems (PAS) [4,6], (B) against local hospital data (registration forms, operation log books, patient records, etc.) [1,7,8,9] and (C) against information obtained from the patients using different types of questionnaires [1].

In order to use PAS data for evaluation of register completeness, the PAS system itself must have high completeness regarding surgeries on a national level, as well as correct procedure coding. However, as the Lithuanian e-health IT system is still in a stage of implementation and has not yet achieved full functioning, the validation of register data is unreliable. Similarly, local hospital data in Lithuania are not in electronic form yet, including operation log books, and the accuracy of medical documentation is of limited value for use in registry validation, as part of the information is also incomplete.

Therefore, we decided that to validate the revision rates in the Lithuanian Arthroplasty Register it would be more reasonable to contact the registered primary total knee arthroplasty (TKA) patients and inquire if they had been subject to additional surgeries on their operated knee within two years. Such a study should reflect the accuracy of the Lithuanian Arthroplasty Register and validate the reported revisions according our “true revision” definition.

## 2. Materials and Methods

The Lithuanian Arthroplasty Register (LAR) was established in 2011. All orthopedic departments within the country who undertake primary or revision arthroplasty participate. A minimal dataset is reported via the Internet and includes personal identification number, age, sex, side, diagnosis and operative details. In cases of reoperation, the reason for revision and the type of revision is also reported [10]. As with many other registers, the LAR defines revision as a second operation after an arthroplasty in which implant components are exchanged, removed or added.

A total of 4269 primary TKAs registered in the LAR between 1 September 2013, and 1 September 2015, were included. Contact details (telephone numbers) were gathered from the operating hospitals and 2 years after the surgery the patients were contacted by phone to inquire if they had been subject to revision. Twenty-two hospitals in Lithuania performed total knee replacements during this period. Nine small hospitals were not able to provide contact details (telephone numbers) for 533 patients (549 knees). Before contacting the patients, their living status was checked in the Lithuanian National Census Register. Patients registered as alive and with a valid telephone number were contacted. Sixty-seven patients (67 knees) had deceased (data from Lithuanian National Census Register) and a further 438 patients (565 knees) appeared to have a wrong or non-valid telephone number leaving 3031 (3091 knees) patients being contacted. Of those, 262 patients (266 knees) refused to participate in the study which left 2769 responders (2825 knees) (Table 1).

The information from the patients was then cross checked against what had been registered in the LAR, and in case of a revision not having been registered the patient hospital charts were investigated. Thus, the patients were followed up with regarding revision and/or death until 1 September 2017, but not more than two years postoperatively.

Patients claiming to have been subjected to additional surgery on their index knee were asked further about when this occurred, what hospital performed the procedure, as well as the type of procedure. After that, the hospital that the patient claimed to have performed the procedure was asked to provide the relevant medical charts to ascertain that the additional surgery performed was a true revision by the LAR definition.

For a number of small hospitals which were not able to provide contact details, as their patients records were incomplete for those patients, we did a separate analysis in order to evaluate if those registered in the LAR were different from the patients that could be contacted with respect to demographics or number of revision performed. A comparison of the gender distribution and patient age was performed between the study participants and those patients for whom data were not provided from the hospitals.

For descriptive statistics, we used frequencies, means ± standard deviations, ranges and 95% confidence intervals (CI). The cumulative survival rate (CRR) was calculated using Kaplan–Meyer statistics and graphs were plotted with CIs. A *p*-value < 0.05 was considered significant. STATA v13 [11] was used for calculations.

This study was approved by the national ethical committee (No.158200-16-832-371).

## 3. Results

Sixty-one patients said that reoperation had been performed on the index knee within two years of their primary surgery. After checking with the clinics, 51 of these were found to be true revisions, and 10 were surgical procedures on the knee, but not true revisions by the LAR criteria. These included two wound revisions, two periprosthetic fractures, three revisions on the contralateral side, two operations for patella ligament rupture and one for a quadriceps muscle rupture, all which do not affect the implant in the investigated knee (Table 2).

Out of 51 true revisions, we found that 46 were registered to LAR as revised, while five (9.8%) revisions were missing. The missing revisions were: one arthrodesis, one extraction of the implant, one insert exchange and two tibia component exchanges. On the other hand, six patients (six knees) reported that they had not been subject to further surgery although a revision had been registered in the LAR, and this could be confirmed by examination of hospital charts.

Figure 1 shows the calculated cumulative survival rate (CRR) before and after the LAR was been updated with the five missing TKA revisions.

With respect to the patients that could not be contacted, their demographic data and survival rates were similar to that of those that could be contacted (Table 3).

## 4. Discussion

In the development of new implants or fixation methods in arthroplasty surgery, a stepwise introduction [12,13,14] is necessary to reduce the risk of implant failure. Patient related outcomes would be different if implants were introduced in to the market after being tested in clinical trials and register based studies.

The randomized clinical trial is considered the gold standard for the design of clinical research, but such studies are not always possible to conduct on surgical treatments. The main question is not the efficacy of knee replacement, but how various knee prostheses or surgical techniques compare with each other. These parameters are suitable for a randomization. However, there are several drawbacks to such a study design, such as the risk of performance bias between centers of excellence and routine surgery [15,16,17]. In addition, a randomized trial is expensive to perform and results are received late due to long-term follow-up. While register based studies usually include a much higher number of patients, bias between centers is avoided, inferior performance of implant or surgical technique is detected immediately and finally, such studies are less expensive. However, to provide high quality data and recommendations from register based studies high completeness of input data is essential, making validation studies important.

The validation process of the LAR resulted in five TKA revisions not reported to the register being discovered using information from the patients and confirmation by a check of the hospital records. The proportion of unreported TKA revisions was 9% during the inclusion period, which we consider as acceptable as the Lithuanian Arthroplasty Register is relatively young, being established in 2011. For comparison, the Swedish Knee Arthroplasty Register reported 20% of knee revisions as missing during 1975–1995 [18], the Dutch register reported 10% of revisions as missing in 2013 [19] and the Columbian Institutional Arthroplasty Register found 9.2% of knee revisions were missing in the period from March to September 2015 [9].

A comparison of the survival curves before and after the registry was updated with the five missed TKA revisions showed only a minor difference in the cumulative survival rate (Figure 1).

The missing revisions were: One arthrodesis, one extraction, one insert exchange and two tibial component exchanges. We assume that some surgeons were still unfamiliar with the definitions of this young register, and did not realize that arthrodesis, component removal or changes of insert are considered revisions requiring reporting to the registry.

The limitation of our study is that out of the 4269 primary TKAs, feedback was received from 2769 TKA patients (2825 knees) providing the information for about 66% of cases from the whole registry. One might suspect that the relatively low response rate might have the same effect on the validation results. However, we performed a comparison analysis of those being contacted and non-responders, aiming to compare their differences in demographics and revision rates reported to the registry. We observed that responder and non-responder groups were very similar in the distribution of age, gender and reported revision rates, and thus we assume that although only 66% of TKA patients were contacted, they probably truly represent the whole group. On the other hand, as revisions are used as the end-point in survival analysis, it is more important that revisions are included for the investigated primary TKA patients.

Another limitation is that we were not able to validate the reporting of primary TKAs using a patient administrative system. However, as long as the underreporting of primaries is not biased, this effect is of less importance than that of missing revisions.

## 5. Conclusions

We conclude that the Lithuanian Arthroplasty Register has an acceptable completeness with respect to TKA revisions as only 9.8% of revisions were underreported in the register.

## Figures and Tables

**Figure 1 medicina-55-00310-f001:**
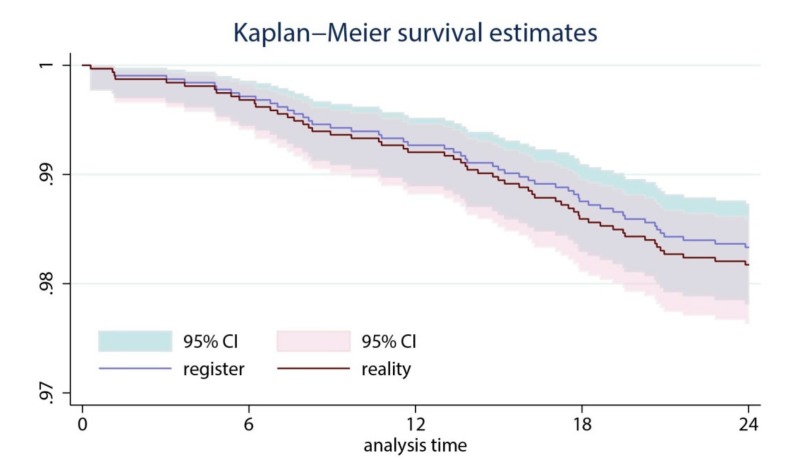
Cumulative survival rate of the investigated total knee arthroplasties (TKA’s) before and after the registry was updated by adding the five non-registered revision TKAs. The shaded area shows the 95% confidence interval.

**Table 1 medicina-55-00310-t001:** Results of questionnaire regarding 4269 knees.

	Patients	Knees
Total number of TKAs registered in LAR during inclusion period	4069	4269
Hospitals were not able to provide contacts of patients	533	549
TKA patients of whom contacts were received	3536	3720
Died before the end of follow up period	67	67
Refused to participate in the study	262	266
Lost (wrong contacts/emigration)	438	562
Agreed to participate in the study (responders)	2769	2825

**Table 2 medicina-55-00310-t002:** The reasons for the 10 not true revisions by our criteria.

Reason	No.
Wound revision	2
Periprosthetic fracture not affecting the implants	2
Revision on the contralateral side	3
Operation for patella ligament rupture	2
Quadriceps muscle rupture	1
Total	10

**Table 3 medicina-55-00310-t003:** Patient data comparing the number of revisions registered in the Lithuanian Arthroplasty Register for participants and non-responders.

Data	Participants, N = 2825 TKA	Hospitals Did Not Provided Data to, N = 549 TKA	Refused to Participate in the Study, N = 266
Age (years), mean ± SD	68 ± 8	68 ± 8	71 ± 8
Gender: F/M	2225/600 (79%/21%)	446/103 (81%/19%)	223/43 (84%/16%)
No. revised	46 (1.6%)	9 (1.6%)	3 (0.9%)

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
