# Peer review of "Validation of Lithuanian Arthroplasty Register Telephone Survey of 2769 Patients Operated for Total Knee Replacement"

_medicina, 2019, doi:10.3390/medicina55060310_

Round 1

Reviewer 1 Report

Thank you for giving me the opportunity to review this manuscript on Lithuanian Arthroplasty Register validation. I believe that this is an interesting study that merits publication after some minor changes. Herein I provide some comments:

TITLE: I would propose the authors to amend their title as follows: “Validation of the Lithuanian Total Knee Arthroplasty Register via a Telephone Survey of 2769 Patients”

ABSTRACT: Please pay attention to data that describes Materials and data that describes results (see comments bellow; RESULTS section). Please explain what “true revisions” means. This section should be clear and attractive for the reader to continue reading your manuscript, so try to provide a short, appealing and straightforward abstract. Omit unsubstantial information as the exact dates, etc.

INTRODUCTION: The last two paragraphs are similar; describing the aim of the study. I would propose the authors to merge them in one, restating their purpose in a more succinct fashion.

MATERIALS AND METHODS: Lines 86-89. Please explain further the “separate analysis”.

RESULTS: Lines 96-102 are not Results but Materials. Pls move this data to the corresponding section. The same should be done to the abstract subsections.

DISCUSSION: This section needs to be expanded. Pls argue further your findings, add more references and explain why your findings are important. Additionally, add a paragraph with the limitations of the study.

Author Response

Response to Reviewer 1 comments

Thank you for giving me the opportunity to review this manuscript on Lithuanian Arthroplasty Register validation. I believe that this is an interesting study that merits publication after some minor changes. Herein I provide some comments:

Point 1: TITLE: I would propose the authors to amend their title as follows: “Validation of the Lithuanian Total Knee Arthroplasty Register via a Telephone Survey of 2769 Patients”

Response 1: Unfortunately we cannot change the title as there is no such organization as Lithuanian Total Knee Arthroplasty.  Lithuanian Arthroplasty register includes total knee replacement, also partial knee replacements (unicondylar knees) and all type of hip replacements. Also our original title includes the information regarding specific patient group being investigated in our validation study.

Point 2:ABSTRACT: Please pay attention to data that describes Materials and data that describes results (see comments bellow; RESULTS section). Please explain what “true revisions” means. This section should be clear and attractive for the reader to continue reading your manuscript, so try to provide a short, appealing and straightforward abstract. Omit unsubstantial information as the exact dates, etc.

Response 2: We now explained the term “true revisions”, the definition has been added in “Materials and methods” section in Abstract.

We now shortened the Abstract, removing exact dates and other unsubstantial information.

Point 3: INTRODUCTION: The last two paragraphs are similar; describing the aim of the study. I would propose the authors to merge them in one, restating their purpose in a more succinct fashion.

Responce 3: We now merged the last two paragraphs in “Introduction” section and reformulated last sentence in “Introduction” section.

Point 4: MATERIALS AND METHODS: Lines 86-89. Please explain further the “separate analysis”.

Responce 4: We now added the information in “Materials and methods” section describing the comparison analysis performed between study participants and patients from non-participating hospitals.

Pont 5: RESULTS: Lines 96-102 are not Results but Materials. Pls move this data to the corresponding section. The same should be done to the abstract subsections.

Response 5: We now moved data from “Results” section to the “Materials and methods” section.

Point 6:  DISCUSSION: This section needs to be expanded. Pls argue further your findings, add more references and explain why your findings are important. Additionally, add a paragraph with the limitations of the study.

Response 6: We now expanded the discussion section; more references have been added and paragraph regarding the limitations of the study is now included.

Reviewer 2 Report

This is an important study for the LAR in order to substantiate the registrations on the registry.

However the methodology and poor patient follow up has resulted in only 66% return rate. In my opinion this is far too low to validate the accuracy of data collection on the registry and brings into question not only the revision data but also the validation of the index procedure data. I don't believe this is scientifically sound and in this form should not be published. 

Perhaps it is publishable in an opinion piece or "letter to the editor" but even in this form it would need considerable discussion on why there is a lack of data and provide remedies to fix this.

Also most other registries would consider periprosthetic fracture a revision procedure.

Author Response

This is an important study for the LAR in order to substantiate the registrations on the registry.

Point 1: However, the methodology and poor patient follow up has resulted in only 66% return rate. In my opinion this is far too low to validate the accuracy of data collection on the registry and brings into question not only the revision data but also the validation of the index procedure data. I don't believe this is scientifically sound and in this form should not be published. 

Response1: We acknowledge that relatively low response rate can be considered as a limitation of the study, however comparison analysis of age and gender distributions and implant survival rates between responders and non-responders showed no significant differences, suggesting that analysis of responders probably represents whole material.

Perhaps it is publishable in an opinion piece or "letter to the editor" but even in this form it would need considerable discussion on why there is a lack of data and provide remedies to fix this.

Point 2: Also most other registries would consider periprosthetic fracture a revision procedure.

Response 2: We disagree that most of other arthroplasty registers consider periprosthetic fractures around the implanted knee prosthesis as revision procedure if osteosynthesis is conducted and no parts of the prosthesis were exchanged in fracture fixation procedure.

Please find bellow the description of definitions of revision (failure) after total knee replacement extracted from various world known Registers latest reports.

Swedish knee Register 2017: “Revision is defined as a new operation in a previously resurfaced knee in which one or more of the components are exchanged, removed or added (incl. arthrodesis or amputation).”

British 2018: „Within the NJR, a revision is defined as any operation in which any prosthesis or part of a prosthesis is either removed, exchanged or inserted for any reason into a joint in which there is an existing joint replacement.

Australian register 2018: „Revision knee replacements are re-operations of previous knee replacements where one or more of the prosthetic components are replaced, removed, or one or more components are added. Revisions include reoperations of primary partial, primary total or previous revision procedures.

Reviewer 3 Report

The authors have successfully validated the data from the Lithuanian Arthroplasty Register and shown that the registry data is robust. The method has been used by other arthroplasty registers.

The result is important if we are to believe that results from this register is credible. The result shows that the register can meet the standards laid down by ISAR.

Author Response

The authors have successfully validated the data from the Lithuanian Arthroplasty Register and shown that the registry data is robust. The method has been used by other arthroplasty registers.

The result is important if we are to believe that results from this register is credible. The result shows that the register can meet the standards laid down by ISAR.

Thank you very much for reviewing our paper.